# Cryo-EM structures of human ZnT8 in both outward- and inward-facing conformations

Jing Xue[1,2], Tian Xie[2,3], Weizhong Zeng[1,2], Youxing Jiang[1,2]*, Xiao-chen Bai[2,3]*

[1]Howard Hughes Medical Institute and Department of Physiology, University of Texas Southwestern Medical Center, Dallas, United States; [2]Department of Biophysics, University of Texas Southwestern Medical Center, Dallas, United States; [3]Department of Cell Biology, University of Texas Southwestern Medical Center, Dallas, United States

**Abstract** ZnT8 is a $Zn^{2+}/H^+$ antiporter that belongs to SLC30 family and plays an essential role in regulating $Zn^{2+}$ accumulation in the insulin secretory granules of pancreatic β cells. However, the $Zn^{2+}/H^+$ exchange mechanism of ZnT8 remains unclear due to the lack of high-resolution structures. Here, we report the cryo-EM structures of human ZnT8 (HsZnT8) in both outward- and inward-facing conformations. HsZnT8 forms a dimeric structure with four $Zn^{2+}$ binding sites within each subunit: a highly conserved primary site in transmembrane domain (TMD) housing the $Zn^{2+}$ substrate; an interfacial site between TMD and C-terminal domain (CTD) that modulates the $Zn^{2+}$ transport activity of HsZnT8; and two adjacent sites buried in the cytosolic domain and chelated by conserved residues from CTD and the His-Cys-His (HCH) motif from the N-terminal segment of the neighboring subunit. A comparison of the outward- and inward-facing structures reveals that the TMD of each HsZnT8 subunit undergoes a large structural rearrangement, allowing for alternating access to the primary $Zn^{2+}$ site during the transport cycle. Collectively, our studies provide the structural insights into the $Zn^{2+}/H^+$ exchange mechanism of HsZnT8.

*For correspondence:
youxing.jiang@utsouthwestern.
edu (YJ);
Xiaochen.Bai@UTSouthwestern.
edu (X-B)

**Competing interests:** The authors declare that no competing interests exist.

## Introduction

$Zn^{2+}$, as a structural and catalytic cofactor, plays crucial roles in regulating protein functions, and thereby is essential for many cellular processes, including cell development and immunological functions (*Maret, 2013*). Not surprisingly, $Zn^{2+}$ deficiency can lead to a variety of disabling human diseases, such as diabetes, cancer and Alzheimer's disease (*Prasad, 2013*). In humans, the intracellular level of zinc is tightly regulated by two major $Zn^{2+}$ transporter families, the SLC39 (ZIPs) family and the SLC30 (ZnTs) family (*Eide, 2006*). ZIPs mediate the $Zn^{2+}$ uptake either from the extracellular space or organelle into cytosolic space, whereas ZnTs transport $Zn^{2+}$ in the opposite direction. ZnT8, one of the 10 ZnTs that have been identified so far (*Chimienti et al., 2004*), is localized in the insulin secretory granules of pancreatic β cells and responsible for the accumulation of high level of zinc inside granules, which is required for the packing of insulin in the hexameric form (*Davidson et al., 2014*). ZnT8 knock-out in mice leads to severe defects in insulin processing, storage and secretion as well as glucose tolerance, indicating its critical role in β-cell function and glucose metabolism (*Pound et al., 2009*; *Wijesekara et al., 2010*). Indeed, the aberrant function of ZnT8 is linked to both type 1 and 2 diabetes (*Chabosseau and Rutter, 2016*). ZnT8 has been shown to be one of the major autoantigens in type 1 diabetes (*Wenzlau et al., 2007*), and autoantibodies against ZnT8 (ZnT8A) can be detected in ~70% of young patients with type 1 diabetes (*Kawasaki, 2012*). In addition, an R325W mutation of ZnT8 has been shown to increase the risk of type 2 diabetes (*Scott et al., 2007*; *Sladek et al., 2007*). On the other hand, in a 2014 study,

researchers also identified 12 rare loss-of-function mutations caused by truncation in ZnT8 that can lead to a 65% reduction in the risk of type 2 diabetes (*Flannick et al., 2014*). Therefore, ZnT8 has become an important target for diagnosis, prevention or intervention of both of type 1 and 2 diabetes.

ZnT8 functions as a $Zn^{2+}/H^+$ exchanger (*Ohana et al., 2009*; *Shusterman et al., 2014*). Despite of extensive biochemical and physiological studies of ZnT8, our understanding of its $Zn^{2+}/H^+$ exchange mechanism is hindered by the lack of high-resolution structures of ZnT8 in different transporting states. To date, the structures of YiiP, a bacterial member of SLC30 family that is distantly related to human ZnT8 with 14% sequence identity (*Figure 1—figure supplement 1*), have been determined by X-ray crystallography and cryo-electron microscopy (cryo-EM) (*Coudray et al., 2013*; *Lopez-Redondo et al., 2018*; *Lu et al., 2009*; *Lu and Fu, 2007*), and these structures have been used as the model for ZnT8. The crystal structure of *Escherichia coli* YiiP (EcYiiP) was determined in an outward-facing conformation at 2.9 Å resolution and the transporter forms a 'V'-shaped homo-dimer, with each subunit consisting of a transmembrane domain (TMD) with six membrane-spanning helices and a cytosolic C-terminal domain (CTD) (*Lu et al., 2009*; *Lu and Fu, 2007*). Distinct from the crystal structure of EcYiiP, the cryo-EM structure of *Shewanella oneidensis* YiiP (SoYiiP) was determined in an inward-facing conformation at 4.1 Å resolution (*Coudray et al., 2013*; *Lopez-Redondo et al., 2018*). Comparison of the two structures reveals major conformational changes in the TMD. Since the two structures of YiiP were determined by different methods with the proteins stabilized in different chemical environments (lipid vs detergent), it is unclear whether the observed structural differences truly reflect the conformational changes upon $Zn^{2+}$ transport. In light of their low-sequence similarity (*Figure 1—figure supplement 1*), it is also unclear whether human ZnT8 (HsZnT8) shares the same $Zn^{2+}$ transport mechanism as bacterial YiiP. Here, we determine the cryo-EM structures of HsZnT8 in both outward (luminal)- and inward (cytosolic)-facing states. These structures reveal a large number of novel features in HsZnT8 that are distinct from bacterial YiiP. Structural comparison between the two states elucidates the conformational rearrangements at the TMD of each ZnT8 subunit during $Zn^{2+}$ transport cycle. Our structural results, together with mutagenesis and cell based functional assays, illustrate a plausible $Zn^{2+}/H^+$ exchange mechanism in ZnT8 and also provide a structure platform for diabetes drug development targeting HsZnT8.

## Results

### Structure determination of HsZnT8

HsZnT8 isoform B lacking the N-terminal 49 residues gave higher protein expression than the full-length isoform A and was used for structure determination in this study (*Figure 1—figure supplement 2a*). To be consistent with the amino acid numbering used for isoform A in other studies, the starting residue of isoform B is numbered as 50 in the subsequent description of our structural and functional analyses. The purified HsZnT8 in digitonin detergent formed a dimer in solution and eluted as a monodisperse peak on size exclusion chromatography (*Figure 1—figure supplement 2b*). The small size of HsZnT8 (about 70 kDa as a dimer) imposed a high technical challenge for its structure determination using cryo-EM. Volta phase plate was employed in data collection to enhance the contrast for better image alignment (*Danev and Baumeister, 2016*), and the initial data was collected using the wild-type HsZnT8 (HsZnT8-WT) sample prepared in the presence of $Zn^{2+}$ (1 mM). The single particle reconstruction yielded a map of the transporter in an outward-facing conformation, but the overall resolution of 4.1 Å is insufficient for accurate model building (*Figure 2—figure supplement 1*). Subsequently, we introduced two mutations, D110N and D224N, at the predicted primary $Zn^{2+}$ binding site within the TMD of HsZnT8 (HsZnT8-DM) (*Hoch et al., 2012*). As the primary site functions as the transfer center for $Zn^{2+}$ transport, we reasoned that by abolishing its $Zn^{2+}$-binding capability, the HsZnT8-DM mutant could potentially stabilize the transporter in a specific state. Indeed, the double mutations appeared to further stabilize the transporter in outward-facing conformation and data collected using HsZnT8-DM sample prepared in the absence of $Zn^{2+}$ yielded a map of the outward-facing transporter with an improved resolution (~3.8 Å) as compared to HsZnT8-WT, allowing us to build a near complete model containing 302 out of 320 residues for each subunit, with the help of the crystal structure of EcYiiP (*Lu et al., 2009*; *Figure 1—figure supplements 3*, *4a*).

Interestingly, the EM data collected with HsZnT8-WT in the absence of $Zn^{2+}$ yielded a structure of the dimeric transporter with heterogeneous conformations- one subunit in an inward-facing and the other in an outward-facing conformation (*Figure 3—figure supplement 1*). Similar heterogeneous conformations have been observed in other multimeric transporters, such as nucleoside (*Hirschi et al., 2017*) and glutamate transporters (*Arkhipova et al., 2020*). Albeit the cryo-EM structure of HsZnT8-WT in the absence of $Zn^{2+}$ was determined at a low overall resolution (5.9 Å), the secondary structure features were well resolved for both TMD and CTD, enabling us to build a relatively accurate model for the inward-facing subunit by rigid-body fitting of the individual transmembrane helix and CTD from the outward-facing structure into the map (*Figure 3—figure supplement 1g*). Furthermore, the density for most of the linkers between adjacent transmembrane helices were well defined in the cryo-EM map, further facilitating the model building of the inward-facing structure (*Figure 3—figure supplement 1g*).

## Overall structure of HsZnT8 in the outward-facing conformation

The structure of HsZnT8 in the outward-facing state shows a 'V'-shaped dimeric architecture similar to the crystal structure of EcYiiP, with the two TMDs forming the branches, whereas the two tightly packed CTDs located at the base of the 'V' and mediating the majority of the dimerization interactions (*Figure 1a–c*, *Figure 1—figure supplement 5a*). The gap between the two TMDs in the HsZnT8 dimer is narrower than that of EcYiiP (*Figure 1—figure supplement 5b*).

The CTD of HsZnT8 adopts a αββαβ fold, a conserved feature of SLC30 family (*Figure 1d*, *Figure 1—figure supplements 2a*, *6*). While one face of the three-stranded β sheet is covered by the two α helices, the other face of the β sheet provides the major contact surface for the dimerization interactions with its counterpart from the neighboring CTD (*Figure 1e*). The protein contacts at the core of this dimer interface are mainly hydrophobic, consisting of multiple aromatic residues including Trp306s, His304s and Phe134s (*Figure 1e*). Notably, Phe134 comes from the extended loop between TM2 and TM3 of TMD, which protrudes into the CTD of the neighboring subunit and tucks into the pocket between β1 and the C-terminal tail. The CTD dimerization also involves multiple hydrogen bonding interactions, for example the H-bonds between Arg331s and Gln350s (*Figure 1e*).

Arg325, whose mutation (R325W) is associated with higher risk of type 2 diabetes, is located in a short loop (residues 321–327) between β2 and α2 of CTD. This loop has also been proposed to be the antigen epitope for autoantibodies from type 1 diabetes patients. Structurally, the two epitope loops in HsZnT8 dimer are physically separated from each other and completely exposed to the cytosolic solution at the distal tip of CTD, providing ample space for antibody access (*Figure 1e*). On the contrary, the two equivalent loops in EcYiiP dimer are tightly packed with each other and contribute to part of the dimerization contact (*Figure 1—figure supplement 5d*). As the epitope loop is remote from the $Zn^{2+}$ transport pathway, it is unclear how R325W mutation and the binding of autoantibodies to this loop affect the transporter function.

Among the six transmembrane helices (TM1-TM6) in the TMD of each HsZnT8 subunit, TMs 3 and 6 also participate in the dimerization through hydrophobic interactions with their counter parts from the neighboring subunit at the cytosolic leaflet of the membrane (*Figure 1f*). In addition, Arg138s at the N-terminal ends of TM3s and Glu276s at the C-terminal ends of TM6s from both subunits converge at the cytosolic surface of the membrane and form a cluster of salt bridges (*Figure 1e*). The extensive dimerization interactions at CTDs as well as TMs 3 and 6 imply that these parts of the protein likely remain static during $Zn^{2+}$ transport cycle, leaving other part of the TMD (TMs 1, 2, 4 and 5) undergo conformational changes as will be further discussed later.

## $Zn^{2+}$ binding sites in CTD

The structure of HsZnT8-DM reveals two adjacent $Zn^{2+}$ binding sites (designated as $S_{CD1}$ and $S_{CD2}$) within each CTD, in a pocket encircled by the U-shaped loop near the C-terminus of the protein right after β3 (*Figure 2a–c*). The EM map shows strong density peaks at both $S_{CD1}$ and $S_{CD2}$ that can be assigned to $Zn^{2+}$ ions (*Figure 2b,c*). Although $Zn^{2+}$ binding was also observed in the CTD of EcYiiP, the exact locations and the surrounding chemical environments of $S_{CD1}$ and $S_{CD2}$ in HsZnT8 are distinct from those in EcYiiP (*Figure 1—figure supplement 5d*). In HsZnT8, $Zn^{2+}$ ions in both $S_{CD1}$ and $S_{CD2}$ are coordinated in a classical tetrahedral geometry with the $S_{CD1}$ $Zn^{2+}$ chelated by His52,

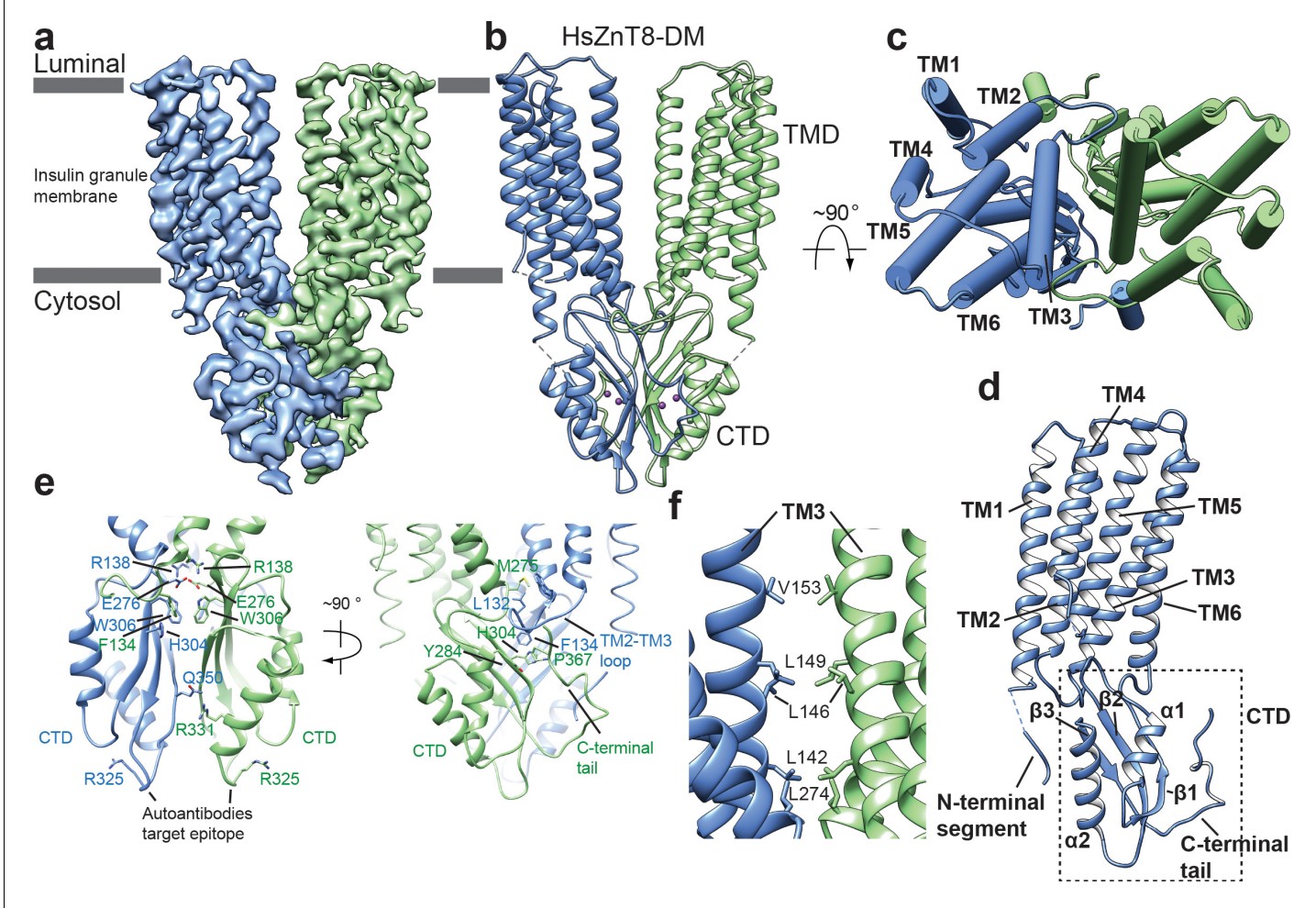

**Figure 1.** Overall structure of HsZnT8-DM in the outward-facing conformation. (**a**) Cryo-EM reconstruction of HsZnT8-DM in the outward-facing conformation (contour level: 0.018, 4.8 σ). The two subunits are colored in blue and green, respectively. (**b**) The ribbon representation of the HsZnT8-DM dimer in the side view. (**c**) The cylinder representation of the HsZnT8-DM dimer in the top view with TMs from one subunit labeled. (**d**) The ribbon representation of a single HsZnT8-DM subunit with secondary structural elements labeled. (**e**) Zoomed-in views of the interface between the two adjacent CTDs in two orthogonal orientations. (**f**) Zoomed-in view of the interface between the two adjacent TMDs.

The online version of this article includes the following figure supplement(s) for figure 1:

**Figure supplement 1.** Sequence alignment of HsZnT8 (Q8IWU4), EcYiiP (P69380) and SoYiiP (Q8E919).

**Figure supplement 2.** Purification of the HsZnT8.

**Figure supplement 3.** Cryo-EM analysis of the HsZnT8-DM in the absence of $Zn^{2+}$.

**Figure supplement 4.** Cryo-EM density of HsZnT8-DM in the absence of $Zn^{2+}$.

**Figure supplement 5.** Structural comparison between HsZnT8 and EcYiiP in outward-facing conformation.

**Figure supplement 6.** Sequence alignment of ZnT8 from human (Q8IWU4), mouse (Q8BGG0), western clawed frog (Q5XHB4), chicken (A0A1D5NY81) and zebrafish (A0A0R4IFM6).

His54, Cys361, and Cys364, and the $S_{CD2}$ $Zn^{2+}$ chelated by Cys53, His301, His318 and Glu352 (*Figure 2b,c*). Consistent with our model, Cys361 and Cys364 at the C-terminus of ZnT8 have been predicted, based on sequence alignment, to be important for zinc binding (*Parsons et al., 2018*). Notably, His52, Cys53 and His54 come from the HCH (His52-Cys53-His54) motif at the N-terminus of the neighboring HsZnT8 subunit. The HCH motif seals off $S_{CD1}$ and $S_{CD2}$, making the two bound $Zn^{2+}$ ions buried in the protein. As the HsZnT8-DM sample was prepared in the absence of $Zn^{2+}$, the bound $Zn^{2+}$ ions likely come from the host cells for protein expression and appear to be integral part of the protein with high-affinity binding. The HCH motif is highly conserved in ZnT8s (*Figure 1— figure supplement 6*) as well as some subfamilies of ZnT, but is not present in YiiP transporter

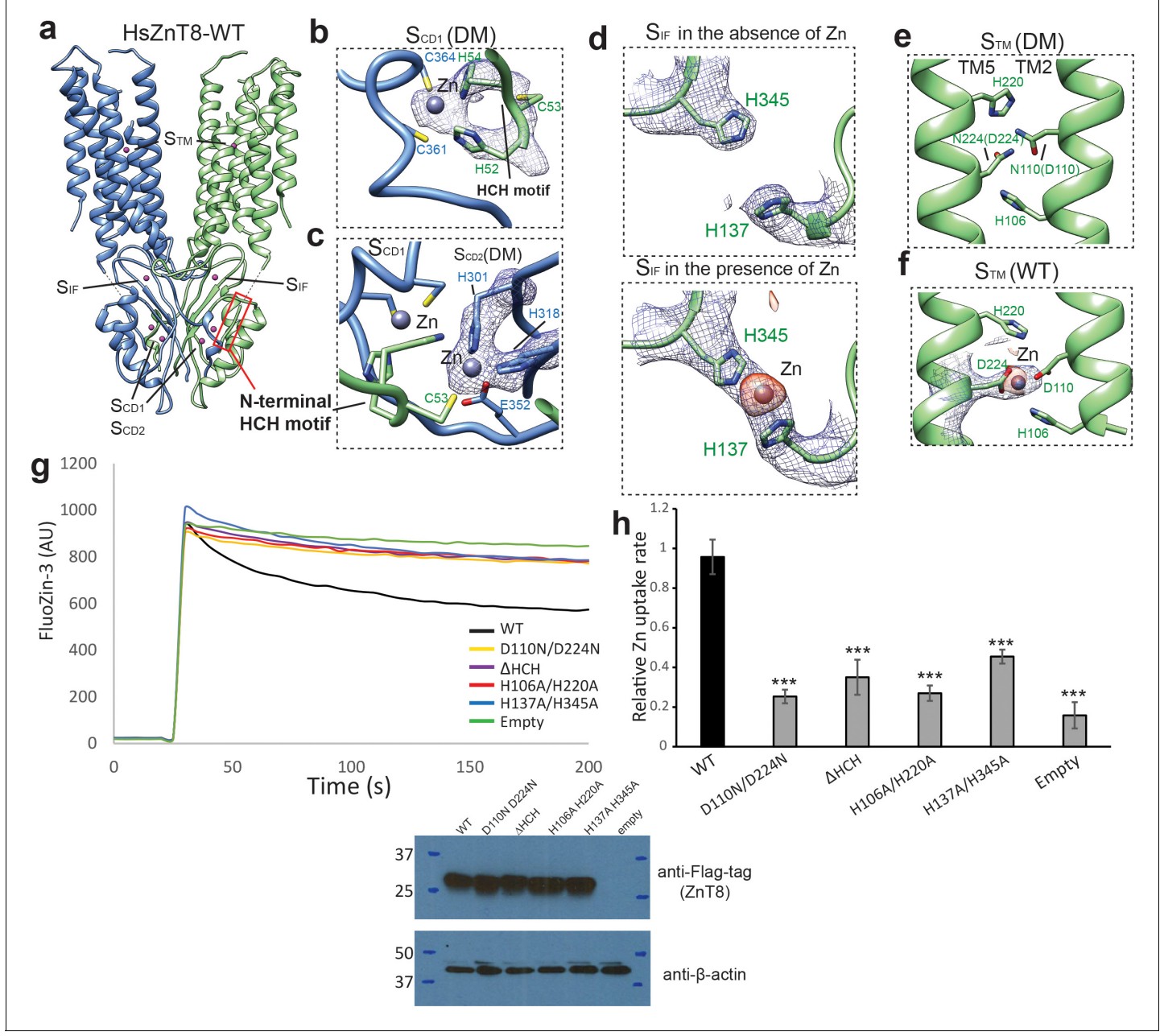

**Figure 2.** The $Zn^{2+}$ binding sites in HsZnT8. (**a**) Overall view of $Zn^{2+}$ binding sites in HsZnT8-WT. The location of each site is indicated by a dash box. (**b**) and (**c**) Detailed views of $S_{CD1}$ and $S_{CD2}$ in the structure of HsZnT8-DM in the absence of $Zn^{2+}$. The cryo-EM densities are displayed as blue mesh (6 σ). (**d**), (**e**) and (**f**) Detailed views of $S_{IF}$ and $S_{TM}$ in the structures of HsZnT8-DM in the absence of $Zn^{2+}$ and HsZnT8-WT in the presence of $Zn^{2+}$. The cryo-EM densities are displayed as blue mesh (6 σ). The density differences between the cryo-EM maps of HsZnT8-DM in the absence of $Zn^{2+}$ and HsZnT8-WT in the presence of $Zn^{2+}$ are shown as orange blobs (8 σ). (**g**) Representative fluorescence traces of vesicular $Zn^{2+}$ uptake for the cells expressing wild-type HsZnT8 or its mutants. Non-induced HEK293F cells (empty) was used as the negative control. Expression levels of HsZnT8 wild type and mutants were monitored by western blot using anti-flag antibody. Mouse anti-β-actin was used as a loading control. (**h**) Relative $Zn^{2+}$ uptake rate between HsZnT8 and its mutants (Mean ± SD). The uptake rates were normalized to the expression levels of HsZnT8. Each experiment was repeated four times. Significance calculated using two-tailed students t-test; between wild-type and mutants; ***p<0.001.

The online version of this article includes the following source data and figure supplement(s) for figure 2:

**Source data 1.** Source data for *Figure 2g*.
**Source data 2.** Source data for *Figure 2h*.
**Figure supplement 1.** Cryo-EM analysis of the HsZnT8-WT in the presence of $Zn^{2+}$.

whose N-terminus does not participate in $Zn^{2+}$ binding (*Figure 1—figure supplement 1*). As a result, the two CTD $Zn^{2+}$ ions in YiiP are partially solvent-exposed with water molecules participating in the ion coordination (*Lu et al., 2009*).

The HCH motif is connected to the N-terminus of TM1 by a short 7-residue linker whose density is clearly visible in the low-pass filtered cryo-EM map (*Figure 1—figure supplement 4b*), but cannot be modeled due to the lack of side-chain densities. The $Zn^{2+}$-mediated tight tethering between HCH motif and the neighboring CTD likely imposes constrain to the position and movement of TM1 (*Figures 1b* and *2a*). It is, therefore, appealing to speculate that the $Zn^{2+}$-mediated stretching and tethering of the N-terminal segment of HsZnT8 that immediately precedes TM1 may play a critical role in the stability and transport function of the TMD in HsZnT8. To test this hypothesis, we generated an N-terminal deletion mutation lacking the HCH motif (ΔHCH) and tested the mutant function using a cell-based $Zn^{2+}$ uptake assay. With comparable protein expression level, cells expressing ΔHCH mutant showed marked reduction in $Zn^{2+}$ uptake activity as compared to cells expressing wild-type transporter (*Figure 2g,h*), confirming the functional importance of the HCH motif.

From the structure of HsZnT8-WT determined in the presence of $Zn^{2+}$, we also observed cryo-EM density from a third $Zn^{2+}$ ion coordinated by His137 and His345 at the interface between CTD and TMD, which is designated as $S_{IF}$ (*Figure 2a,d*). This $Zn^{2+}$ density was not present in the HsZnT8-DM structure which was determined in the absence $Zn^{2+}$ (*Figure 2d*), suggesting a lower affinity $Zn^{2+}$ binding at $S_{IF}$ as compared to the other two CTD sites ($S_{CD1}$ and $S_{CD2}$). Interestingly, $S_{IF}$ is positioned right at the cytosolic entrance to the $Zn^{2+}$ transport pathway (*Figure 2a*) and mutation of the two $S_{IF}$ histidines resulted in marked reduction in $Zn^{2+}$ transport rate (*Figure 2g,h*). As the physiological function of ZnT8 is to transport $Zn^{2+}$ ions from low concentration cytosol (free $Zn^{2+}$ concentration:~1 nM) to high concentration granule (free $Zn^{2+}$ concentration:~120 nM) (*Krezel and Maret, 2006*; *Hessels et al., 2015*), we suspect $S_{IF}$ can help trap $Zn^{2+}$ ions and effectively increase the local $Zn^{2+}$ concentration, which in turn facilitate the ion transport against over 100-fold concentration gradient.

## $Zn^{2+}$ binding sites in TMD

The primary $Zn^{2+}$ binding site in HsZnT8 TMD (designated as $S_{TM}$) is highly conserved in SLC30 family (*Figure 1—figure supplement 6*). $S_{TM}$ is surrounded by His106, Asp110, His220 and Asp224 at the middle of the membrane and exposed to the luminal solution in the outward-facing structure (*Figure 2e*). The density of a bound $Zn^{2+}$ can be defined in the EM map of HsZnT8-WT (*Figure 2f*, *Figure 2—figure supplement 1*). Although the side-chains of those ligand residues are not well defined in the map due to the resolution limit, their locations and conservation likely point to a classical tetrahedral coordination as observed in EcYiiP structure. As expected, $S_{TM}$ is vacant in HsZnT8-DM structure because the two $Zn^{2+}$-coordinating acidic residues (Asp110 and Asp224) were replaced with asparagines in the mutant and the sample was also prepared in the absence of $Zn^{2+}$ (*Figure 2e*). Although the structure of HsZnT8-DM is virtually identical to that of HsZnT8-WT (*Figure 2—figure supplement 1e*), the mutant has almost no $Zn^{2+}$ transport activity (*Figure 2g,h*), confirming the central role of $S_{TM}$ to HsZnT8 function.

## Structural changes from outward- to inward-facing states

Intriguingly, the two subunits in the cryo-EM structure of HsZnT8-WT obtained in the absence of $Zn^{2+}$ adopt different conformations, with one subunit in inward-facing but the other in outward-facing conformation (*Figure 3a*). Although at lower resolution, the secondary structure features from almost all part of the inward-facing subunit can be well defined, allowing us to visualize the structural changes between inward and outward-facing conformations (*Figure 3a,b*, *Figure 3—figure supplement 1*). In the outward-facing conformation, TMs 1–5 are tightly bundled at the cytosolic half of TMD, occluding the space between $S_{TM}$ and the cytosol; at the luminal half of TMD, TMs 2–6 enclose a deep cavity, penetrating into the central $S_{TM}$ where $Zn^{2+}$ binds and providing a solvent accessible passageway from luminal side (*Figure 2a*, *Figure 3c*, *Figure 3—figure supplement 2d*). In the inward-facing conformation, TMs 2–6 are tightly bundled at the luminal half of TMD and occlude the solvent accessible passageway from luminal side; TMs 1–6 enclose a wide open cavity at the cytosolic half of TMD, making $S_{TM}$ readily accessible from cytosol (*Figure 3a–c*, *Figure 3—figure supplement 2d*).

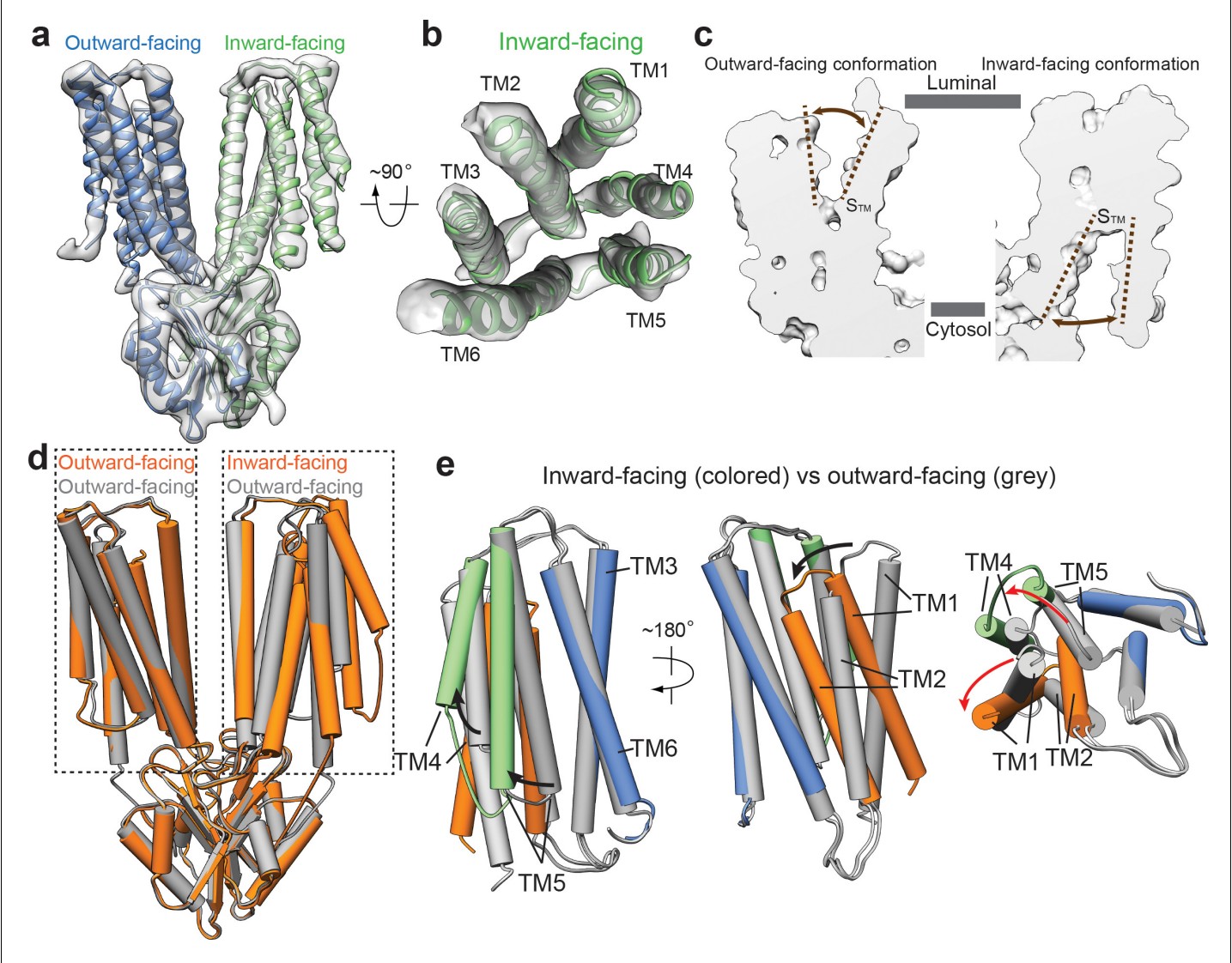

**Figure 3.** Structural transitions between outward- and inward-facing conformations. (a) 3D reconstruction of HsZnT8-WT in the absence of $Zn^{2+}$ and the ribbon representation of the model fitted into cryo-EM map. (b) Top view of the TMD (TMs 1–6) from the inward-facing subunit in ribbon representation fitted into cryo-EM map. (c) The surface-rendered models of an outward-facing subunit from the HsZnT8-DM structure (left) and an inward-facing subunit from the HsZnT8-WT structure obtained in the absence of $Zn^{2+}$ (right). The brown dashed lines indicate the opening of $S_{TM}$ to either the luminal or the cytosolic side. (d) Superposition between the structures of HsZnT8-DM dimer (grey) and HsZnT8-WT dimer (orange) both obtained in the absence of $Zn^{2+}$ (e) Superposition of the TMD structures between the outward-facing subunit from HsZnT8-DM (with TMs colored in grey) and the inward-facing subunit from HsZnT8-WT (with TMs 1–2 colored in brown, TMs 4–5 in green and TMs 3 and six in blue), revealing two different types of rocking motions (indicated by the arrows).

The online version of this article includes the following figure supplement(s) for figure 3:

**Figure supplement 1.** Cryo-EM analysis of the HsZnT8-WT in the absence of $Zn^{2+}$.

**Figure supplement 2.** Structural comparison between HsZnT8 in outward- and inward-facing conformations.

**Figure supplement 3.** Structural comparison between HsZnT8 and SoYiiP in inward-facing conformation.

Significant scissors-like rotation between two TMDs of bacterial YiiP dimer was observed between outward- and inward-facing conformations (*Lopez-Redondo et al., 2018*). Consequently, the relative position between the two TM3 helices changes significantly between the two conformations (*Figure 1—figure supplement 5a*, *Figure 3—figure supplement 3b*). In sharp contrast, the regions that participate in dimerization of HsZnT8, including CTD, TM3 and TM6, can all be well superimposed between the structures of inward and outward-facing conformations, indicating that these regions

remain static during Zn²⁺ transport cycle (*Figure 3d,e*). However, large structural rearrangements occur among TMs 1, 2, 4 and 5 of HsZnT8, resulting from two different types of rocking motions (*Figure 3e*, *Figure 3—figure supplement 2a–c*, *Video 1*). From outward to inward-facing conformation, TMs 4 and 5, as a bundle, swing away from TMs 3 and 6, using their luminal ends as the pivot point; TM5 tilts approximately 35° from the center of TMD to the peripheral, resulting in the opening of cytosolic ion pathway (*Figure 3e*, *Figure 3—figure supplement 2a–c*, *Video 1*). TMs 1 and 2, as another bundle, undergo large inward rotation hinged at the cytosolic end of TM2, occluding the luminal ion pathway (*Figure 3e*, *Figure 3—figure supplement 2a–c*, *Video 1*).

In addition, we superimposed the TMD regions between the inward-facing structures of HsZnT8 and SoYiiP. This comparison revealed no major structural differences for most of TMs between HsZnT8 and SoYiiP, with the exception of TM1 (*Figure 3—figure supplement 3d,e*). In the inward-facing conformation of HsZnT8, TM1 packs tightly against TM2 in a manner similar to that in the outward-facing conformation. TM1 of SoYiiP, however, rotates away from TM2 during the transition from outward- to inward-facing states (*Figure 3—figure supplement 3d*). As mentioned above, the tethering of the N-terminal segment of HsZnT8 to CTD may restrain the flexibility of TM1 and limit its rocking rotation upon conformational change. This explains why the TM1s of HsZnT8 and SoYiiP undergo different types of structural changes during the transport cycle. We also noticed that, HsZnT8 adopts a more open inward-facing conformation as compared to SoYiiP, due to the altered position of TM1 (*Figure 3—figure supplements 2d*, *3d*). It is tempting to speculate that, the wide open cytosolic cavity in the inward-facing conformation of HsZnT8 could facilitate the fast diffusion of Zn²⁺ from the cytosol to the primary Zn²⁺ site in TMD and thereby may increase the efficiency of Zn²⁺ uptake into insulin granule.

## The mechanism for Zn²⁺/H⁺ antiport in HsZnT8

The conformational heterogeneity of HsZnT8-WT in the absence of Zn²⁺ along with small size of the protein likely causes difficulty in obtaining a high-resolution cryo-EM structure. However, the observation of different conformations between the two subunits within each HsZnT8-WT dimer implies that the Zn²⁺ transport activity of HsZnT8 is unlikely to be coupled between the two subunits within a dimer, allowing each subunit to carry out Zn²⁺ transport activity independently (*Figure 4a*). No occluded state has been observed in all of our structural analysis, leading us to suggest a simple two-state model for Zn²⁺ transport function of HsZnT8 (*Figure 4b*). Each transporter subunit may simply shuttle between inward- and outward-facing states with $S_{TM}$ housing the Zn²⁺ ion in a pH-dependent manner. At inward-facing state, higher cytosolic pH environment allows for high-affinity Zn²⁺ binding at $S_{TM}$. The presence of a secondary Zn²⁺ site at the cytosolic entrance of the ion pathway may help increase the local Zn²⁺ concentration and facilitate the Zn²⁺ occupancy at $S_{TM}$ even at low concentration. Upon moving to the outward-facing state, $S_{TM}$ becomes exposed to the low pH environment of the insulin secretory granule and the subsequent protonation of $S_{TM}$, likely through the two histidines, effectively reduce the Zn²⁺ affinity and promote its release (*Figure 4b*). The transporter would then move back to inward-facing states where the apo $S_{TM}$ gets deprotonated at higher pH and ready for the next transport cycle (*Figure 4b*). Thus, the direction and efficiency of Zn²⁺ transport are simply determined by the pH and Zn²⁺ concentrations on both sides of the transporter.

## Discussion

In this work, we determined the first cryo-EM structures of HsZnT8 in both outward- and inward-facing conformations. Our structural and functional analyses reveal a total of 4 Zn²⁺ binding sites in each subunit, including one primary site in the TMD ($S_{TM}$), one site at the interface between TMD and CTD ($S_{IF}$), and two adjacent sites located in an open pocket of CTD covered by a HCH motif from the N-terminal segment of neighboring subunit ($S_{CD1}$ and $S_{CD2}$). The primary Zn²⁺ binding site in the TMD of ZnT8

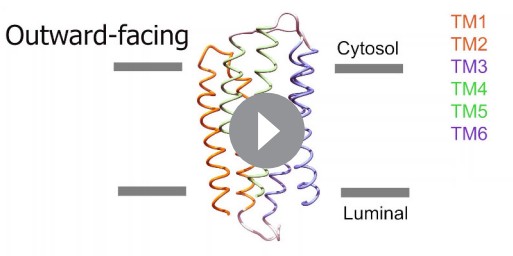

**Video 1.** Structural transitions of the TMD of HsZnT8 between outward- and inward-facing conformations. https://elifesciences.org/articles/58823#video1

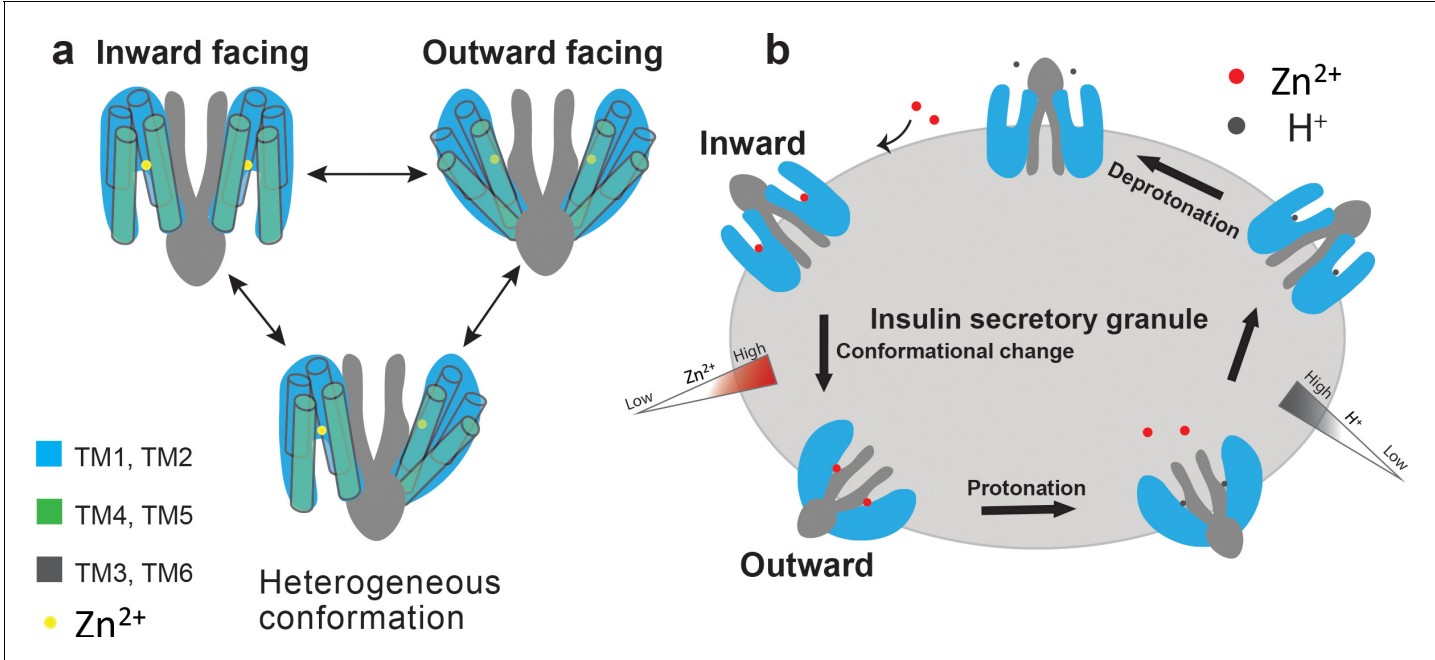

**Figure 4.** Proposed working model of HsZnT8. (**a**) A cartoon representation illustrates that each ZnT8 subunit may shuttle between inward- and outward-facing conformations independently during the $Zn^{2+}$ transport cycle. (**b**) Schematic representation of a simplified model for the transport cycle of ZnT8. For simplicity, both subunits in a ZnT8 dimer are drawn in homogeneous conformations, but they likely carry out $Zn^{2+}/H^+$ exchange function independent from each other, as shown in **a**.

The online version of this article includes the following figure supplement(s) for figure 4:

**Figure supplement 1.** Mapping of the disease-associated point mutations onto HsZnT8.

consists of His-Asp-His-Asp residues, with one residue difference as compared to Asp-Asp-His-Asp of YiiP. It has been shown that the replacement of Asp to His in the primary site of ZnTs significantly increases the specificity of $Zn^{2+}$ binding (*Hoch et al., 2012*). We proposed that the weak binding of $Zn^{2+}$ to the interfacial site $S_{IF}$ in HsZnT8 may help to increase local $Zn^{2+}$ concentration at the entrance of cytosolic ion pathway, which could facilitate the recruitment of cytosolic $Zn^{2+}$ to the primary site in TMD. Indeed, our cell based uptake assay showed that disruption of $S_{IF}$ markedly reduced the transport activity of HsZnT8. This interfacial $Zn^{2+}$ binding site is not present in YiiP.

The two adjacent $Zn^{2+}$ binding sites in CTD, $S_{CD1}$ and $S_{CD2}$, are localized in a similar position between ZnT8 and YiiP, but are formed by very different structural elements. In HsZnT8, $S_{CD1}$ and $S_{CD2}$ is surrounded by the β-sheet and C-terminal tail of CTD from one subunit and the N-terminal segment from the other. Therefore, the binding of $Zn^{2+}$ to $S_{CD1}$ and $S_{CD2}$ in HsZnT8 will anchor each CTD to the TMD of the neighboring subunit. In contrast, both of C-terminal tail of CTD and N-terminal segment are not involved in the formation of $S_{CD1}$ and $S_{CD2}$ in YiiP. Instead, $S_{CD1}$ and $S_{CD2}$ of YiiP is sandwiched between two neighboring CTDs, such that the binding of $Zn^{2+}$ to $S_{CD1}$ and $S_{CD2}$ plays an important role in the dimerization of CTD in YiiP.

A comparison of our outward- and inward-facing structures of HsZnT8 reveals detailed structural rearrangements within each TMD during the transport cycle. Importantly, the motions observed during the transition from outward- to inward-facing states are significantly different between ZnT8 and YiiP (*Figure 3—figure supplement 3*). Firstly, different from YiiP, no inter-TMDs rotation occurs during the transport cycle of ZnT8. The positions of TM3 and 6 of HsZnT8 remain unchanged between outward- and inward-facing states. Secondly, due to the tethering of HCH motif in the N-terminal segment to CTD, the movement of TM1 of HsZnT8 is largely restricted. As a result, TM1 of HsZnT8 undergoes distinct structural rearrangement between outward- and inward-facing states, as compared to that of YiiP, leading to a wider opening of cytosolic cavity in the inward conformation, which might be important for the efficient diffusion of $Zn^{2+}$ from the cytosol to the primary $Zn^{2+}$ binding site in TMD. This observation partially explains why the HCH motif is critical for the transport

activity of HsZnT8. Notably, mutation of a single residue in the HCH motif of ZnT2 (H54R) was directly linked with transient neonatal zinc deficiency (TNZD), further indicating the significance of this motif in the function of SLC30 family of zinc transporter (*Figure 4—figure supplement 1*; *Chowanadisai et al., 2006*). It also worth noting that similar HCH motif has been found in the C-terminus of copper importer Ctr1 that can bind $Cu^{2+}$ and is important for the regulation of Ctr1 activity (*Maryon et al., 2013*).

In conclusion, our structural and functional analyses provide a wealth of information for defining the $Zn^{2+}/H^+$ antiporter mechanism of HsZnT8. Additionally, our detailed structural model of HsZnT8 presented here provide a basis for understanding a number of mutations in ZnT8 that are associated with diabetes (*Figure 4—figure supplement 1*), and may serve as the template for homologous modeling of other members in SLC30 family.

# Materials and methods

## Key resources table

| Reagent type (species) or resource | Designation | Source or reference | Identifiers | Additional information |
|---|---|---|---|---|
| Strain, strain background (*Escherichia coli*) | TOP10 | Thermo Fisher Scientific | Cat# 18258012 | |
| Strain, strain background (*Escherichia coli*) | DH10bac | Thermo Fisher Scientific | Cat# 10361012 | |
| Cell line (*Spodoptera frugiperda*) | Sf9 cells | Thermo Fisher Scientific | Cat# 11496015; RRID:CVCL_0549 | |
| Cell line (*Homo sapiens*) | FreeStyle 293 F cells | Thermo Fisher Scientific | Cat# R79007; RRID:CVCL_D603 | |
| Recombinant DNA reagent | pEZT-BM | DOI:10.1016/j.str.2016.03.004 | Addgene:74099 | |
| Transfected construct (*Homo-sapiens*) | pEZT-BM-ZNT8-N$_{Flag}$ and mutations | This paper | N/A | transfected construct (human) |
| Antibody | Mouse monoclonal anti-FLAG tag | Sigma | Cat# F1804 | WB (1:5000) |
| Antibody | Mouse monoclonal anti-b-actin | Santa Cruz Biotechnology | Cat# sc-69879; RRID:AB_1119529 | WB (1:200) |
| Antibody | Mouse IgG HRP linked whole Ab | GE healthcare | Cat# NA931V; RRID:AB_772210 | WB (1:10000) |
| Chemical compound, drug | Sodium Butyrate | Sigma-Aldrich | Cat# 303410 | |
| Chemical compound, drug | n-Dodecyl-b-Maltopyranoside | Anatrace | Cat# D310, D310s | |
| Chemical compound, drug | Cholesteryl hemisuccinate | Sigma-Aldrich | Cat# C6512 | |
| Chemical compound, drug | Lauryl Maltose Neopentyl Glycol | Anatrace | Cat# NG310 | |
| Chemical compound, drug | Digitonin | Sigma | Cat# D141 | |
| Chemical compound, drug | 1,10-Phenanthroline | Sigma | Cat# 131377 | |
| Chemical compound, drug | FluoZin—3 | Thermo Fisher Scientific | Cat# F24194 | |
| Peptide, recombinant protein | 3X FLAG Peptide | Sigma | Cat# F4799 | |
| Software, algorithm | MotionCor2 | *Zheng et al., 2017* | https://emcore.ucsf.edu/ucsf-software | |

*Continued on next page*

*Continued*

| Reagent type (species) or resource | Designation | Source or reference | Identifiers | Additional information |
|---|---|---|---|---|
| Software, algorithm | GCTF | *Zhang, 2016* | https://www2.mrc-lmb.cam.ac.uk/research/locally-developed-software/zhang-software/#gctf | |
| Software, algorithm | RELION | Scheres, 2012 | http://www2.mrc-lmb.cam.ac.uk/relion | |
| Software, algorithm | Chimera | *Pettersen et al., 2004* | https://www.cgl.ucsf.edu/chimera; RRID:SCR_004097 | |
| Software, algorithm | PyMol | Schrödinger | https://pymol.org/2; RRID:SCR_000305 | |
| Software, algorithm | COOT | *Emsley et al., 2010* | https://www2.mrc-lmb.cam.ac.uk/ personal/pemsley/coot; RRID:SCR_014222 | |
| Software, algorithm | PHENIX | *Adams et al., 2010* | https://www.phenix-online.org | |
| Software, algorithm | GraphPad Prism | GraphPad Software | https://www.graphpad.com/scientific-software/prism | |
| Software, algorithm | OriginPro 8 | OriginLab Corp. | https://www.originlab.com | |
| Other | Superose 6 Increase 10/300 GL | GE Healthcare | Cat# 29091596 | |
| Other | Anti-DYKDDDDK G1 Affinity Resin | GeneScript | Cat# 10362101 | |
| Other | Amicon Ultra-15 Centrifugal Filter Units | Milliporesigma | Cat# UFC9100 | |
| Other | Quantifoil R 1.2/1.3 grid Au300 | quantifoil | Cat# Q37572 | |
| Other | Cellfectin | Invitrogen | Cat# 10362100 | |

## Protein expression and purification

Human ZnT8 cDNA (residues 50–369) was cloned into pEZT vector with an N-terminal Flag tag and heterologously expressed in HEK293F cells (R79007, Thermo Fisher Scientific) using the BacMam system. The baculovirus generated in Sf9 cells (11496015, Thermo Fisher Scientific) was used to infect HEK293F cells at a ratio of 40:1 (cells:virus, v/v) and 10 mM sodium butyrate were added to the cell culture to boost protein expression. 48 hr after infection at 37°C, cells were collected by centrifugation at 4000 g. All purification procedures were carried out at 4°C unless specified otherwise. The cell pellet was re-suspended in Buffer A (25 mM Hepes pH 7.4, 150 mM NaCl) supplemented with protease inhibitors (1 mg/ml each of DNase, pepstatin, leupeptin, and aprotinin and 1 mM PMSF) and homogenized by sonication on ice. HsZnT8 protein was extracted with 1.5% (w/v) n-dodecyl-β-d-maltopyranoside (DDM; Anatrace) and 0.02% (w/v) cholesteryl hemisuccinate (CHS; Sigma-Aldrich) by gentle agitation for 2 hr. After extraction, the supernatant was collected after centrifugation at 40,000 g and incubated with anti-Flag M2 affinity resin by gentle agitation for 1 hr. Then the resin was collected on a disposable gravity column and washed with 20 column volume of Buffer A supplemented with 0.05% (w/v) lauryl maltose neopentyl glycol (MNG, Anatrace). HsZnT8 was eluted in Buffer A with 0.05% (w/v) MNG and 0.1 mg/ml Flag peptide. The protein eluate was concentrated and further purified by size-exclusion chromatography on a Superose6 10/300 GL column (GE Healthcare) in Buffer A with 0.06% (w/v) Digitonin. The peak fractions were collected and concentrated to 4 mg/ml for grid preparation.

The HsZnT8 D110N/D224N double mutant construct were generated using QuikChange (Agilent). The same procedure was used to express and purify HsZnT8-DM for cryo-EM analysis.

## Cryo-EM data collection

Purified HsZnT8-DM and HsZnT8-WT either in the presence or absence of $Zn^{2+}$ (1 mM) at 4 mg/ml was applied to a glow-discharged Quantifoil R1.2/1.3 300-mesh gold holey carbon grid (Quantifoil, Micro Tools GmbH, Germany), blotted under 100% humidity at 4°C and plunged into liquid ethane using a Mark IV Vitrobot (FEI). Micrographs were acquired on a Titan Krios microscope (FEI) with a K3 Summit direct electron detector (Gatan) in the super-resolution counting mode, operated at 300 kV using the SerialEM software (*Mastronarde, 2005*). The slit width of the GIF-Quantum energy filter was set to 20 eV. A Volta phase plate was used to enhance low-resolution features (*Danev and Baumeister, 2016*). Micrographs were dose-fractioned into 32 frames at the dose rate of ~2 e$^-$/Å$^2$/frame.

## Image processing and 3D reconstruction

Movie frames of HsZnT8-DM micrographs were motion-corrected and binned two-fold, resulting in the pixel size of 0.83 Å, and dose-weighted using the Motioncorr2 program (*Figure 1—figure supplement 3*; *Zheng et al., 2017*). CTF correction were performed using the GCTF programs (*Zhang, 2016*). The rest of the image processing steps was carried out using RELION 3 (*Zivanov et al., 2018*). A few micrographs from the HsZnT8-DM dataset were used for manual picking of ~1000 particles. These particles were subjected to 2D classification. Class averages representing projections of the HsZnT8 dimer in different orientations were used as templates for automated particle picking from the full datasets. A total of 1,287,890 particles were picked from 3384 micrographs. Particles were extracted and binned by four times (leading to 3.32 Å/pixel) and subjected to 2D classification. Particles in good 2D classes were chosen (569,565 in total) for 3D classification using an initial model generated from a subset of the particles in RELION. Particles from the 3D classes showing good secondary structural features were selected and re-extracted into the original pixel size of 0.83 Å. 3D refinements with C2 symmetry imposed resulted in 3D reconstructions to 4 Å resolution. To improve the resolution, we performed another round of 3D classification by using local search in combination with small angular sampling, resulting a new class showing improved density for the entire protein. The final reconstruction was resolved at overall 3.8 Å resolution.

A total of 1,537,280 particles were picked from 3776 micrographs of HsZnT8-WT in the absence of $Zn^{2+}$ (*Figure 2—figure supplement 1*). 831,115 particles were selected by 2D classification. A map of the ZnT8 in the outward-facing conformation was low-pass filtered and used as the initial reference. The subsequent 3D classification revealed one good class with each ZnT8 subunit adopting a distinct conformation. Therefore, the following 3D refinement was performed with C1 symmetry imposed, leading to a final reconstruction at 5.9 Å resolution.

The same image processing procedure was used to obtain the 3D reconstruction of HsZnT8 in the presence of $Zn^{2+}$, yielding a map at 4.1 Å resolution (*Figure 3—figure supplement 1*). Local resolution was calculated in RELION. Resolution was estimated by applying a soft mask around the protein density with the Fourier Shell Correlation (FSC) 0.143 criterion.

To calculate the difference map, the cryo-EM map of HsZnT8-DM in the absence of $Zn^{2+}$ at 3.8 Å resolution was firstly normalized to the same grey scale as the cryo-EM map of HsZnT8-WT in the presence of $Zn^{2+}$, by using the command 'vop scale' in UCSF Chimera (*Pettersen et al., 2004*). After aligning the two maps together in UCSF Chimera, the difference map was calculated by using the command 'vop subtract'.

## Model building, refinement and validation

Density maps of the HsZnT8-DM in the outward-facing conformation was of sufficient quality for de novo model building in Coot (*Figure 1—figure supplement 4*; *Emsley et al., 2010*), facilitated by previous crystal structure of EcYiiP (PDB:3H90) (*Lu et al., 2009*). The model was manually adjusted in Coot and refined against the map by using the real space refinement module with secondary structure and non-crystallographic symmetry restraints in the Phenix package (*Adams et al., 2010*). The same procedure was used to build the model of HsZnT8-WT in the presence of $Zn^{2+}$.

The density of the HsZnT8-WT in the absence of $Zn^{2+}$ in the inward-facing conformation is relatively poor. The models of CTD and each TM from the outward-facing structure were rigid-body fitted into the cryo-EM density in Coot with good agreement. The model was subsequently refined against the map by using strong secondary structure restraints in Phenix. Model geometries were

assessed by using Molprobity as a part of the Phenix validation tools and summarized in *Supplementary file 1* (*Chen et al., 2010*). The solvent accessible cavities were calculated with the program Caver (*Chovancova et al., 2012*).

The multiple sequence alignments were performed using the program Clustal Omega (*Sievers et al., 2011*).

### Vesicular zinc uptake assay

In a previous study, it has been shown that HsZnT8 expressed in HEK293 cells facilitated vesicular $Zn^{2+}$ uptake when $Zn^{2+}$ was introduced into cytosol by permeabilizing surface membrane using a $Zn^{2+}$ ionophore (*Merriman et al., 2016*). In that study, the vesicular $Zn^{2+}$ accumulation was monitored by Zinpyr-1, a membrane-permeable $Zn^{2+}$-selective fluorescent indicator that can be localized in vesicles. We adopted similar cell-based vesicular $Zn^{2+}$ uptake assay in our study, except that we permeabilized the cells with digitonin and directly monitored the cytosolic $Zn^{2+}$ concentration decrease caused by vesicular $Zn^{2+}$ uptake using $Zn^{2+}$ indicator FluoZin-3 as described below. HEK293F cells expressing HsZnT8 were harvested 36 hr after virus infection. To remove residual media and $Zn^{2+}$, the cell pellet from 30 mL suspension culture was washed with 30 mL of uptake buffer (20 mM Hepes pH7.4 125 mM KCl, 5 mM NaCl, 10 mM Glucose, 10 μM Phenanthroline) three times, and then re-suspended in uptake buffer to a final concentration of $15 \times 10^6$ cells/mL. The cells were maintained on ice throughout. To measure $Zn^{2+}$ uptake, 100 μL of cell suspension was added into each well of a 96-well plate (Corning) and the following reagents were added sequentially: 1 μM FluoZin-3, 0.01% digitonin, and 9 μM $ZnCl_2$. Zinc uptake was monitored by measuring FluoZin-3 fluorescence change for 5 min using a Molecular Devices SpectraMax M3 plate reader (excitation/emission: 490 nm/525 nm).

Western blot analysis was performed on each sample used for uptake assay to ensure protein expression was comparable among the HsZnT8 WT and mutants. Mouse anti-Flag antibodies (Sigma) were used to detect HsZnT8. Mouse anti-β-actin (Santa Cruz Biotechnology) was used as a loading control. HRP-conjugated sheep anti-mouse antibody (GE Healthcare) was used as the secondary antibody. The expression level of HsZnT8 was calculated based on the quantification of the total level of ZnT8 and β-actin from the western blot analysis. All functional data were analyzed in Graph-Pad Prism 8 (GraphPad Software, Inc) or OriginPro (OriginLab Corp.). To obtain the rate of zinc uptake, the linear phase of the uptake measurement – the first 30 s of the reaction following zinc addition – was determined by fitting the data to a linear regression equation in OriginPro 8; the slope of the fit is taken as the rate of zinc uptake. The zinc uptake rates were normalized to the expression level of HsZnT8. Further analysis are indicated in the figure legend.

The identities of all the cell lines have been authenticated. The mycoplasma contamination testing was performed and shown to be negative.

### Quantification and statistical analysis

The number of independent experiment, the method used in statistical test, and the statistical significance are indicated in each figure legend and source manuscript files.

## Acknowledgements

Cryo-EM data were collected at the University of Texas Southwestern Medical Center Cryo-EM Facility, which is funded by the CPRIT Core Facility Support Award RP170644; we thank D Stoddard for technical support and facility access. This work was supported in part by the Howard Hughes Medical Institute (to YJ), the NIH (grant GM079179 to YJ and R01GM136976 to X-CB), the Welch Foundation (grant I-1578 to YJ and I-1944 to X-CB), the Cancer Prevention and Research Institute of Texas (RP160082 to X-CB), and the Virginia Murchison Linthicum Scholar in Medical Research fund (to X-CB).

## Additional information

### Funding

| Funder | Grant reference number | Author |
|--------|------------------------|--------|
| National Institute of General Medical Sciences | R01GM136976 | Xiao-chen Bai |
| Welch Foundation | I-1944 | Xiao-chen Bai |
| Cancer Prevention and Research Institute of Texas | RP160082 | Xiao-chen Bai |
| Howard Hughes Medical Institute | | Youxing Jiang |
| National Institute of General Medical Sciences | GM079179 | Youxing Jiang |
| Welch Foundation | I-1578 | Youxing Jiang |
| Virginia Murchison Linthicum Scholar in Medical Research | | Xiao-chen Bai |

The funders had no role in study design, data collection and interpretation, or the decision to submit the work for publication.

### Author contributions

Jing Xue, Tian Xie, Data curation, Formal analysis, Writing - original draft, Writing - review and editing; Weizhong Zeng, Data curation, Formal analysis; Youxing Jiang, Conceptualization, Formal analysis, Supervision, Funding acquisition, Validation, Writing - original draft, Writing - review and editing; Xiao-chen Bai, Conceptualization, Data curation, Formal analysis, Supervision, Funding acquisition, Validation, Writing - original draft, Writing - review and editing

### Author ORCIDs

Jing Xue (iD) https://orcid.org/0000-0002-7331-1382
Tian Xie (iD) https://orcid.org/0000-0002-5445-664X
Youxing Jiang (iD) https://orcid.org/0000-0002-1874-0504
Xiao-chen Bai (iD) https://orcid.org/0000-0002-4234-5686

### Decision letter and Author response

Decision letter https://doi.org/10.7554/eLife.58823.sa1
Author response https://doi.org/10.7554/eLife.58823.sa2

## Additional files

### Supplementary files

- Supplementary file 1. CryoEM data collection and model statistics.
- Transparent reporting form

### Data availability

The Cryo-EM maps of HsZnT8 determined in three different conditions have been deposited in the Electron Microscopy Data Bank, and the corresponding atomic coordinates have been deposited to the RCSB Protein Data Bank, with the entry ID: EMD-22285 and PDB 6XPD for the structure of HsZnT8-DM, EMD-22286 and PDB 6XPE for the structure of HsZnT8-WT in the presence of zinc, and EMD-22287 and PDB 6XPF for the structure of HsZnT8-WT in the absence of zinc. Source data files have been provided for Figure 2g and 2h.

The following datasets were generated:

| | **Database and** |
|--|--|

| Author(s) | Year | Dataset title | Dataset URL | Identifier |
|---|---|---|---|---|
| Xue J, Xie T, Zeng W, Jiang Y, Bai Xc | 2020 | Cryo-EM structure of human ZnT8 double mutant - D110N and D224N, determined in outward-facing conformation | https://www.ebi.ac.uk/pdbe/entry/emdb/EMD-22285 | Electron Microscopy Data Bank, EMD-22285 |
| Xue J, Xie T, Zeng W, Jiang Y, Bai Xc | 2020 | Cryo-EM structure of human ZnT8 WT, in the presence of zinc, determined in outward-facing conformation | https://www.ebi.ac.uk/pdbe/entry/emdb/EMD-22286 | Electron Microscopy Data Bank, EMD-22286 |
| Xue J, Xie T, Zeng W, Jiang Y, Bai Xc | 2020 | Cryo-EM structure of human ZnT8 WT, in the absence of zinc, determined in heterogeneous conformations- one subunit in an inward-facing and the other in an outward-facing conformation | https://www.ebi.ac.uk/pdbe/entry/emdb/EMD-22287 | Electron Microscopy Data Bank, EMD-22287 |
| Xue J, Xie T, Zeng W, Jiang Y, Bai Xc | 2020 | Cryo-EM structure of human ZnT8 double mutant - D110N and D224N, determined in outward-facing conformation | https://www.rcsb.org/structure/6xpd | RCSB Protein Data Bank, 6xpd |
| Xue J, Xie T, Zeng W, Jiang Y, Bai Xc | 2020 | Cryo-EM structure of human ZnT8 WT, in the presence of zinc, determined in outward-facing conformation | https://www.rcsb.org/structure/6xpe | RCSB Protein Data Bank, 6xpe |
| Xue J, Xie T, Zeng W, Jiang Y, Bai Xc | 2020 | Cryo-EM structure of human ZnT8 WT, in the absence of zinc, determined in heterogeneous conformations- one subunit in an inward-facing and the other in an outward-facing conformation | https://www.rcsb.org/structure/6xpf | RCSB Protein Data Bank, 6xpf |

The following previously published datasets were used:

| Author(s) | Year | Dataset title | Dataset URL | Database and Identifier |
|---|---|---|---|---|
| Lu M, Chai J, Fu D | 2009 | Structural basis for the autoregulation of the zinc transporter YiiP | https://www.ebi.ac.uk/pdbe/entry/pdb/3h90 | Electron Microscopy Data Bank, 3h90 |
| Lopez-Redondo ML, Coudray N, Zhang Z, Alexopoulos J, Stokes DL | 2018 | CryoEM Structure of the Zinc Transporter YiiP from helical crystals | https://www.ebi.ac.uk/pdbe/entry/pdb/5vrf | Electron Microscopy Data Bank, 5vrf |

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
