## [Decision Letter]

**Acceptance summary:**

The authors report several cryo-EM structures of human zinc transporter ZnT8, which uses the transmembrane proton gradient to pump zinc into insulin secretory granules. There, zinc is necessary to pack insulin into secretion-ready hexameric form. The structures push the technical capabilities of cryo-EM because of the comparatively small size of the transporter. They reveal the transport mechanism, the location of mutations associated with increased risk of type 2 diabetes, and the location of the loop that has been proposed to be the target of autoantibodies in type 1 diabetes patients.

**Decision letter after peer review:**

Thank you for submitting your article "Cryo-EM structures of human ZnT8 in both outward- and inward-facing conformations" for consideration by *eLife*. Your article has been reviewed by Olga Boudker as the Senior and Reviewing Editor, and two reviewers. The following individuals involved in review of your submission have agreed to reveal their identity: Aravind Penmatsa (Reviewer #1).

The reviewers have discussed the reviews with one another and the Reviewing Editor has drafted this decision to help you prepare a revised submission.

Summary:

The manuscript reports cryo-EM structures of the dimeric human ZnT8 from the ZIP protein family in the outward-facing conformation at 3.8 Å and inward-facing conformation at a moderate resolution of 5.9 Å. The authors manage to reconstruct the structure of this sub-100kDa (~70kDa) Zn transporter using the phase plate and the study joins a small but increasing list of integral membrane transporter structures recently elucidated through cryo-EM. The biomedical importance of ZnT8 in insulin secretion, Zn enrichment in secretory granules, and elucidation of different conformations along the transport cycle make this study highly relevant. However, more rigorous data analysis and presentation are required to warrant publication. Below are the points raised by the reviewers that would need to be addressed.

Essential revisions:

1) The major concern is with the certainty that the authors claim the presence/absence of Zn^2+^, especially since the wild-type structures solved with and without Zn^2+^ are at only ~4 Å and ~6 Å resolutions.

Throughout the entire manuscript and figures, there are no indications of statistical significance for interpreting cryo-EM densities. It is important that the displayed/compared volumes show parts of the map that have a corresponding level of significance. This is especially concerning the presence and absence of the Zn^2+^ ion density (Figure 2). In making direct comparisons between density maps that contain different statistical compositions and distributions it is essential to scale and normalize the maps with respect to the lowest quality map. Then displaying the normalized data and stating a number of sigmas above the mean threshold to which the density is present, would allow for evaluation of the density maps on an equal footing. This should be explicitly stated in Materials and methods section and figure legends should this measure have been carried out. Moreover, a density difference map should be calculated to adequately satisfy the author's statements of 'unequivocal' and 'clear' presence of Zn^2+^ (subsections “Zn^2+^ binding sites in CTD” and “Zn^2+^ binding sites in TMD”). However, even at ~4 Å and ~6 Å resolutions, there is still a potential for the presence of noise to be misinterpreted as Zn^2+^ density, and I suggest the authors rephrase their strong assertions of modeling ions at this resolution.

2) Subsection “Structural determination of HsZnT8”: building a relatively accurate model at 5.9 Å for the HsZnT8-WT in the absence of Zn^2+^ is quite a bit of a stretch, and side chains cannot accurately be modeled at this resolution. Interpretations should be limited to what one can observe in the density map and leave speculation for the discussion. Moreover, the model for this structure should be deposited as C-α representations, since minimal side-chain densities are observed.

3) HsZnT8-DM in the absence of Zn^2+^ (Figure 1—figure supplement 2), there appears to be quite a bit of variation in the central TMs (TM6 and TM3) after initial 3D refinement. Some explanation should be provided for this, given that this domain is proposed to be static (subsection “Structural changes from outward- to inward-facing states”). Additionally, were the other classes refined? If so, to what resolutions? Even at 6 Å, as in HsZnT8-WT, some interpretations could expand the understanding of the protein dynamics.

4) The data is presented in a way that it is very difficult to assess the statement "the two conformations appear to be energetically equivalent, allowing the transporter to be evenly distributed between the two conformations." (See subsection “The mechanism for Zn^2+^/H^+^ antiporting in HsZnT8”). Consider revising or provide more substantial evidence for the energetics and "even distribution".

5) The authors should include additional discussion of what is known about the Zn^2+^ binding affinity for each of the sites and physiologically relevant available Zn^2+^ concentrations? Would the transporters ever see an environment in which there is no Zn^2+^ present?

6) The authors have proposed that the structures will provide insights into human disease, but fail to follow through on this. Can disease-causing mutations be mapped to the structures generated in this study or homology models created for this purpose? This would elevate your claim and provide substantial advancement in understanding diabetes.

7) Figure 2 legend should be re-written for clarity as it does not specifically indicate which structure/condition is presented. The threshold for evidence of density should be presented and translated into a number of sigmas above the mean.

8) Subsection “Overall structure of HsZnT8 in the outward-facing conformation”: The statement, "Notably, clear lipid density is visible in the space between the two TMDs" should be supported with a figure displaying the densities as well as further discussion on its identity.

9) In the Zn^2+^ uptake experiments (subsection “Zn^2+^ binding sites in CTD”), it is unclear how the data was normalized. The particulars of the data processing were omitted in the Materials and methods section, directing the reader to the figure legend (subsection “Vesicular zinc uptake assay”), where there is no indication of how data was normalized to protein expression.

10) Subsection “Structural determination of HsZnT8”: The authors state HsZnT8-WT in the absence of Zn^2+^ yielded heterogeneous conformations however, Figure 3—figure supplement 1 does not depict this.

11) How do the solvent-accessible transport pathways appear as a 3D volume in the inward- and outward-facing structures?

12) Subsection “Model building, refinement and validation”: Is the modeling really de novo if a template from the crystal structure was used? This should be rephrased to better represent the contribution of previous structural information.

13) Upon examining the coordinates of the three deposited structures it is evident that in the WT HsZnT8 reported without Zinc, the authors model D110 and D224 as asparagines in chain B at site STM. This is a likely modeling error of the authors as it is fairly clear that the authors are reporting HsZnT8 structure without Zinc in the WT state and not as a heterodimer of HsZnT8DM (CH B) and WT HsZnT8 (CH A).

14) Among the residues involved in the charge interlock (R138; E276) between TMs 3 and 6 there is a break in the symmetry between the residues in the WT ZnT8 structure lacking Zinc. The E276 of chain A is positioned away from forming a salt bridge interaction in the structure. Given the poor resolution of this structure, the side chain density is likely not visible in this case. However, given the lack of any structural changes in the cytosolic domains the authors may correct the orientation of this side chain.

15) The H106 which should be coordinating with Zn in the WT structure is modeled away from the Zinc. The imidazole ring should be brought closer within the coordinating radius of the STM Zn ion in the WT-Zn bound structure.

16) The structural comparisons between outward and inward-facing conformations in Figure 3 would be best be represented with the addition of interhelical angles to represent structural transitions particularly between TMs 1, 2, 4 and 5 which undergo substantial changes.

17) None of the figures showing structural overlaps report C-α RMSD values, be it within ZnT8 structure or the comparison between nZnT8 and EcYiiP/SoYiiP structures. The authors could even consider including a table with domain wise RMSD comparisons between different structures.

18) Despite drawing an extensive comparison between the EcYiiP and SoYiip with HsZnT8, the MSA in Figure 1—figure supplement 5 does not report structure-based sequence comparison between HsZnT8 vs. the prokaryotic Zn transporters. Having this multiple sequence alignment would help the readers significantly.

19) The authors do not particularly indicate the concentrations of Zn used in their work. It is not clear from the methods or the text as to what is the concentration of zinc when they say 'presence of Zn'.

20) The Zn uptake assay displaying activity is a fairly qualitative one and does not report any kinetic parameters. Would the authors be able to provide KM and Vmax values for the transport and their mutants as part of the revision?

21) The discussion of the putative Zn-binding site at the luminal face of the transporter as potential 'luminal Zn sensor' is speculative at best since the earlier citation by Gola et al., 2019, suggests that there is a loss of transport activity upon mutations of E88 and D103. While this suggests that the residues are important for Zn permeation, it does not indicate as to why the site could sense Zn. Also, if the site senses Zn at high concentrations the transport activity is likely to be retained except at high luminal Zn levels. The authors may consider altering this part of the Discussion section.

22) The dual conformation found in the WT ZnT8 structure in the absence of Zinc is a rather interesting observation. While it is plausible that the transporter has a two-state model without a likely occluded state it would be interesting for the authors to discuss the possibility of the two protomers, transporting Zn, to be decoupled from the dimerization helices and the CTD and transport Zn independent of each other.

23) The summary figure (Figure 4) can be improved by adding the pH gradients and Zn concentration gradients across the bilayer. The release of Zn inside the secretory granule can be directly depicted as being released from the molecule upon protonation rather than as arrows between two states.

[Editors' note: further revisions were suggested prior to acceptance, as described below.]

Thank you for resubmitting your work entitled "Cryo-EM structures of human ZnT8 in both outward- and inward-facing conformations" for further consideration by *eLife*. Your revised article has been evaluated by Olga Boudker as Senior and Reviewing Editor.

The manuscript has been improved but there are some remaining issues that need to be addressed before acceptance, as outlined below:

As you will see from the comments below, the reviewers are questioning the assignment of the excess density as a lipid. You should substantiate the assignment and edit the text to reflect uncertainty.

Reviewer #1:

The authors have carefully addressed the concerns that I had raised in the earlier version and have made changes both to the text and the figures as recommended. The manuscript figures display marked improvement and provide greater clarity to the reader in this version. I do not have any major concerns and the manuscript can be accepted after a couple of minor concerns are addressed.

Reviewer #2:

The authors have addressed most of the points that were raised in the initial review. I do appreciate the authors' responses, and new analyses and discussions added to the revised manuscript. However, the manuscript still lacks a satisfactory discussion related to the lipids identified in the in HsZnT8 structure. Specific points are given below and need to be addressed.

- Subsection “Overall structure of HsZnT8 in the outward-facing conformation”: "Notably, clear lipid density is visible in the space between the two TMDs". While the authors have provided a new figure displaying anticipated lipid density (Figure 1—figure supplement 3C), it remains unclear why these densities were ascribed to lipid molecules. The data presented does not support the identification of a lipid. There is only a small density, no bulky head group and no continuous density to lipid bilayer boundaries. What evidence do you have to support ascribing density as lipid? This density could be due to image processing artifacts, noise, bound detergent, etc. It would be impactful to provide clarification/discussion on identity of lipid-like densities or remove this supposition from the manuscript.

- Figure 1A legend – the authors should indicate a threshold level used to display the EM density map; if the map was filtered for this display, it should be also indicated in the figure legend.

- Figure 1—figure supplement 3 – "Lipid densities…." – please provide statistical definition of visualized lipid-like densities. Why these densities were escribed to lipid molecules? It would be impactful to provide some clarification.

- Figure 1—figure supplement 5 and Figure 1—figure supplement 6. Indicate GI/protein accession numbers for the sequence alignments. Indicate how the multiple alignments were accomplished.

- Figure 2B,C,D: "The densities for the Zn^2+^ ion are displayed as blue mesh (6 σ)…" – Cryo-EM densities shown as a blue mesh at 6σ include protein as well. Consider revising.

- Figure 2G and Figure 3C – The surface-rendered model of the outward-facing (Figure 2) and inward-facing (Figure 3) subunit from the HsZnT8-WT structure obtained in the absence of Zn. It would make better sense to show both conformations side-by-side in the same figure.

- Figure 3—figure supplement 1B – The ResMap data could be presented for additional clarity by using an alternate coloring scheme or the same scale as in Figure 1—figure supplement 2.

- While the % particles in each image processing step used to determine the final map were included as previously requested, the % particles in the discarded maps from 3D classification and 3D refinement will provide additional clarity for data distribution.

- Indicate if RMSDs are based on C-α or all-atom.

---

## [Author Response]

Essential revisions:1) The major concern is with the certainty that the authors claim the presence/absence of Zn^2+^, especially since the wild-type structures solved with and without Zn^2+^ are at only ~4 Å and ~6 Å resolutions.Throughout the entire manuscript and figures, there are no indications of statistical significance for interpreting cryo-EM densities. It is important that the displayed/compared volumes show parts of the map that have a corresponding level of significance. This is especially concerning the presence and absence of the Zn^2+^ ion density (Figure 2). In making direct comparisons between density maps that contain different statistical compositions and distributions it is essential to scale and normalize the maps with respect to the lowest quality map. Then displaying the normalized data and stating a number of sigmas above the mean threshold to which the density is present, would allow for evaluation of the density maps on an equal footing. This should be explicitly stated in Materials and methods section and figure legends should this measure have been carried out. Moreover, a density difference map should be calculated to adequately satisfy the author's statements of 'unequivocal' and 'clear' presence of Zn^2+^ (subsections “Zn^2+^ binding sites in CTD” and “Zn^2+^ binding sites in TMD”). However, even at ~4 Å and ~6 Å resolutions, there is still a potential for the presence of noise to be misinterpreted as Zn^2+^ density, and I suggest the authors rephrase their strong assertions of modeling ions at this resolution.

Thank this reviewer for this critical comment. We have normalized the cryo-EM map of HsZnT8-DM at 3.8 Å resolution to the same grey scale as the cryo-EM map of HsZnT8-WT in the presence of Zn^2+^ at 4.1 Å resolution, by using the command “vop scale” in Chimera. We have stated the number of sigmas to which the densities of Zn^2+^ are displayed in the figure legend of Figure 2. In addition, we calculated the density difference between the normalized cryo-EM maps of HsZnT8 in the absence and presence of Zn^2+^. This difference map is presented as orange blobs in Figure 2. Following reviewer’s comment, we also removed all the strong assertions, such as “unequivocal”, from the text. The detailed approaches for normalizing the cryo-EM map and calculating the difference map are described in the Materials and methods section.

2) Subsection “Structural determination of HsZnT8”: building a relatively accurate model at 5.9 Å for the HsZnT8-WT in the absence of Zn^2+^ is quite a bit of a stretch, and side chains cannot accurately be modeled at this resolution. Interpretations should be limited to what one can observe in the density map and leave speculation for the discussion. Moreover, the model for this structure should be deposited as C-α representations, since minimal side-chain densities are observed.

We agree with the reviewer that the side chains of the protein cannot be accurately modeled at this resolution. Therefore, the conformational changes of ZnT8 TMD between outward- to inward-facing states are only depicted and discussed at the secondary structure level throughout the entire manuscript. We have removed all the side-chains from the model of HsZnT8-WT in the absence of Zn^2+^.

3) HsZnT8-DM in the absence of Zn^2+^ (Figure 1—figure supplement 2), there appears to be quite a bit of variation in the central TMs (TM6 and TM3) after initial 3D refinement. Some explanation should be provided for this, given that this domain is proposed to be static (subsection “Structural changes from outward- to inward-facing states”). Additionally, were the other classes refined? If so, to what resolutions? Even at 6 Å, as in HsZnT8-WT, some interpretations could expand the understanding of the protein dynamics.

Thank this reviewer for raising this question. After we align all classes shown in Figure 1—figure supplement 2 together, all the TMs between different classes can be perfectly overlapped with each other (see Author response image 1), suggesting that there are no conformational variations in TMs among all these classes. It has been well known that the extreme hydrophobic air-water interface on the cryo-EM grids may potentially damage protein complex. It is possible that the particles from other “bad” classes may be damaged more than those selected from this single “good” class, which could explain why the 3D reconstruction for other classes were determined in similar conformations but at lower resolution.

**Author response image 1. sa2fig1:** 

4) The data is presented in a way that it is very difficult to assess the statement "the two conformations appear to be energetically equivalent, allowing the transporter to be evenly distributed between the two conformations." (See subsection “The mechanism for Zn^2+^/H^+^ antiporting in HsZnT8”). Consider revising or provide more substantial evidence for the energetics and "even distribution".

Thank this review for the critical comment. Indeed, we don’t know for certain that the outward- and inward-facing conformations are energetically equivalent in HsZnT8. We also can’t provide more evidence to support this statement. We, therefore, remove this statement from the revised manuscript.

5) The authors should include additional discussion of what is known about the Zn^2+^ binding affinity for each of the sites and physiologically relevant available Zn^2+^ concentrations? Would the transporters ever see an environment in which there is no Zn^2+^ present?

We can’t find any literature reporting the Zn^2+^ binding affinities of ZnTs. As there are totally 4 different Zn^2+^ binding sites in each subunit of ZnT8, it would be difficult to measure the Zn^2+^ binding affinity for each site individually.

The free Zn^2+^ concentration in cytosol is between 200 pM to 1.5 nM; whereas the free Zn^2+^ concentration in insulin secretory granule is about 120 nM. Thus, there are always Zn^2+^ present on both cytosolic and luminal sides of ZnT8, although the Zn^2+^ concentration difference between the two sides are more than 100 fold. We have added this information in the revised manuscript.

6) The authors have proposed that the structures will provide insights into human disease, but fail to follow through on this. Can disease-causing mutations be mapped to the structures generated in this study or homology models created for this purpose? This would elevate your claim and provide substantial advancement in understanding diabetes.

We thank the reviewer for the suggestion. We have prepared a new figure supplement (Figure 4—figure supplement 1) to map the disease-associated point mutations of ZnT8 as well as those mutations found in other ZnTs but are conserved in ZnT8 onto the structure.

7) Figure 2 legend should be re-written for clarity as it does not specifically indicate which structure/condition is presented. The threshold for evidence of density should be presented and translated into a number of sigmas above the mean.

We have rewritten the legend of Figure 2 to improve its clarity, and stated the sigma level of the Zn^2+^ density in the legend.

8) Subsection “Overall structure of HsZnT8 in the outward-facing conformation”: The statement, "Notably, clear lipid density is visible in the space between the two TMDs" should be supported with a figure displaying the densities as well as further discussion on its identity.

We have prepared a new figure supplement (Figure 1—figure supplement 3C) to show the lipid density between the two TMDs.

9) In the Zn^2+^ uptake experiments (subsection “Zn^2+^ binding sites in CTD”), it is unclear how the data was normalized. The particulars of the data processing were omitted in the Materials and methods section, directing the reader to the figure legend (subsection “Vesicular zinc uptake assay”), where there is no indication of how data was normalized to protein expression.

Thank this reviewer for this critical comment. We have repeated all the uptake assays. The results of western blot analysis of HsZnT8 WT and mutants are shown in Figure 2H, indicating the protein expression level in each uptake assay. β-actin was used as the loading control. The relative expression level for each HsZnT8 sample was calculated through the quantification of the western blotting results of HsZnT8 and β-actin, by using ImageJ. Subsequently, the relative uptake rate for each HsZnT8 sample was calculated from the vesicular zinc uptake assay (Figure 2H and more detail in method section). Finally, the uptake rates were normalized to the HsZnT8 expression levels, and are shown in Figure 2I. We have rewritten the legend and method in the revised manuscript.

10) Subsection “Structural determination of HsZnT8”: The authors state HsZnT8-WT in the absence of Zn^2+^ yielded heterogeneous conformations however, Figure 3—figure supplement 1 does not depict this.

We have labeled the map in Figure 3—figure supplement 1 to indicate the heterogeneous conformations.

11) How do the solvent-accessible transport pathways appear as a 3D volume in the inward- and outward-facing structures?

As suggested by the reviewer, we have prepared a new figure supplement (Figure 3—figure supplement 3D) to show the transport pathways in both outward- and inward-facing conformations as 3D volumes. The 3D transport pathways are calculated by using program Caver.

12) Subsection “Model building, refinement and validation”: Is the modeling really de novo if a template from the crystal structure was used? This should be rephrased to better represent the contribution of previous structural information.

Indeed, we used the structure of EcYiiP as the reference to model HsZnT8. We have rephrased a sentence in the revised manuscript to “allowing us to build a near complete model containing 302 out of 320 residues for each subunit, with the help of the crystal structure of EcYiiP”.

13) Upon examining the coordinates of the three deposited structures it is evident that in the WT HsZnT8 reported without Zinc, the authors model D110 and D224 as asparagines in chain B at site STM. This is a likely modeling error of the authors as it is fairly clear that the authors are reporting HsZnT8 structure without Zinc in the WT state and not as a heterodimer of HsZnT8DM (CH B) and WT HsZnT8 (CH A).

We apologize for this error and corrected it in the revised model.

14) Among the residues involved in the charge interlock (R138; E276) between TMs 3 and 6 there is a break in the symmetry between the residues in the WT ZnT8 structure lacking Zinc. The E276 of chain A is positioned away from forming a salt bridge interaction in the structure. Given the poor resolution of this structure, the side chain density is likely not visible in this case. However, given the lack of any structural changes in the cytosolic domains the authors may correct the orientation of this side chain.

As also suggested by another reviewer, we have removed all the side-chains from the model of HsZnT8 in the absence of Zn^2+^ due to the resolution of the structure. This could avoid any misunderstanding of our structures.

15) The H106 which should be coordinating with Zn in the WT structure is modeled away from the Zinc. The imidazole ring should be brought closer within the coordinating radius of the STM Zn ion in the WT-Zn bound structure.

We have adjusted the side-chain rotamer of H106 in the revised model.

16) The structural comparisons between outward and inward-facing conformations in figure 3 would be best be represented with the addition of interhelical angles to represent structural transitions particularly between TMs 1, 2, 4 & 5 which undergo substantial changes.

As suggested, we have prepared a new figure supplement (Figure 3—figure supplement 3A-B) to depict angular variation of each TM between outward- and inward-facing conformations. We also calculated the interhelical angles of each TM between two states. This result was shown in Figure 3—figure supplement 3C.

17) None of the figures showing structural overlaps report C-α RMSD values, be it within ZnT8 structure or the comparison between nZnT8 and EcYiiP/SoYiiP structures. The authors could even consider including a table with domain wise RMSD comparisons between different structures.

As suggested, we have calculated the RMSD for each domain between HsZnT8 and EcYiiP/SoYiiP. These results are shown in Figure 1—figure supplement 4F and Figure 3—figure supplement 2E.

18) Despite drawing an extensive comparison between the EcYiiP and SoYiip with HsZnT8, the MSA in Figure 1—figure supplement 5 does not report structure-based sequence comparison between HsZnT8 vs. the prokaryotic Zn transporters. Having this multiple sequence alignment would help the readers significantly.

We have prepared a new figure supplement (Figure 1—figure supplement 6) to show the multiple sequence alignment between HsZnT8, EcYiiP and SoYiiP.

19) The authors do not particularly indicate the concentrations of Zn used in their work. It is not clear from the methods or the text as to what is the concentration of zinc when they say 'presence of Zn'.

We have explicitly indicated the Zn^2+^ concentration used in this work in the revised manuscript.

20) The Zn uptake assay displaying activity is a fairly qualitative one and does not report any kinetic parameters. Would the authors be able to provide KM and Vmax values for the transport and their mutants as part of the revision?

The uptake assay used in this study only aims to provide qualitative comparison of the transporter activity between WT and mutants. In our assay, the concentration of contaminated zinc, zinc chelator 1,10-phenanthroline and added zinc are all in the μM range, so we can’t calculate the free zinc concentration accurately. This means we cannot use this assay to accurately measure the KM and Vmax.

21) The discussion of the putative Zn-binding site at the luminal face of the transporter as potential 'luminal Zn sensor' is speculative at best since the earlier citation by Gola et al., 2019, suggests that there is a loss of transport activity upon mutations of E88 and D103. While this suggests that the residues are important for Zn permeation, it does not indicate as to why the site could sense Zn. Also, if the site senses Zn at high concentrations the transport activity is likely to be retained except at high luminal Zn levels. The authors may consider altering this part of the Discussion section.

We agree with this reviewer that the discussion for this putative Zn-binding site is very speculative, and the previously published results on ZnT2 can’t support our speculation. We have completely removed this part in the revised manuscript.

22) The dual conformation found in the WT ZnT8 structure in the absence of Zinc is a rather interesting observation. While it is plausible that the transporter has a two-state model without a likely occluded state it would be interesting for the authors to discuss the possibility of the two protomers, transporting Zn, to be decoupled from the dimerization helices and the CTD and transport Zn independent of each other.

We agree with the reviewer and this is actually what we have suggested in the manuscript, which stated: “the Zn^2+^ transport activity of HsZnT8 is unlikely to be coupled between the two subunits within a dimer, allowing each subunit to carry out Zn^2+^ transport activity independently”.

23) The summary figure (Figure 4) can be improved by adding the pH gradients and Zn concentration gradients across the bilayer. The release of Zn inside the secretory granule can be directly depicted as being released from the molecule upon protonation rather than as arrows between two states.

We have revised Figure 4, as suggested.

[Editors' note: further revisions were suggested prior to acceptance, as described below.]

Reviewer #2:The authors have addressed most of the points that were raised in the initial review. I do appreciate the authors' responses, and new analyses and discussions added to the revised manuscript. However, the manuscript still lacks a satisfactory discussion related to the lipids identified in the in HsZnT8 structure. Specific points are given below and need to be addressed.- Subsection “Overall structure of HsZnT8 in the outward-facing conformation”: "Notably, clear lipid density is visible in the space between the two TMDs". While the authors have provided a new figure displaying anticipated lipid density (Figure 1—figure supplement 3C), it remains unclear why these densities were ascribed to lipid molecules. The data presented does not support the identification of a lipid. There is only a small density, no bulky head group and no continuous density to lipid bilayer boundaries. What evidence do you have to support ascribing density as lipid? This density could be due to image processing artifacts, noise, bound detergent, etc. It would be impactful to provide clarification/discussion on identity of lipid-like densities or remove this supposition from the manuscript.

Thanks to this reviewer for the critical comment. We agree with this reviewer that these two density blobs, shown in Figure 1—figure supplement 3C, could be lipid, detergent, or even noise. Due to the uncertainty in assigning this density, we have completely removed this supposition and Figure 1—figure supplement 3C in the revised manuscript.

- Figure 1A legend – the authors should indicate a threshold level used to display the EM density map; if the map was filtered for this display, it should be also indicated in the figure legend.

We have added the map threshold level in the legend of Figure 1A, as suggested.

- Figure 1—figure supplements 3 – "Lipid densities…." – please provide statistical definition of visualized lipid-like densities. Why these densities were escribed to lipid molecules? It would be impactful to provide some clarification.

We have removed the discussion of this unidentified density as well as Figure 1—figure supplement 3C in the revised manuscript.

- Figure 1—figure supplement 5 and Figure 1—figure supplement 6. Indicate GI/protein accession numbers for the sequence alignments. Indicate how the multiple alignments were accomplished.

As suggested, we have added the protein accession numbers in the legends of Figure 1—figure supplement 5 and Figure 1—figure supplement 6. The multiple sequence alignments were calculated by Clustal Omega. We have indicated this in the Materials and methods section, and cited the original publication of Clustal Omega.

- Figure 2B,C,D: "The densities for the Zn^2+^ ion are displayed as blue mesh (6 σ)…" – Cryo-EM densities shown as a blue mesh at 6σ include protein as well. Consider revising.

We have rewritten this sentence to “The cryo-EM densities are displayed as blue mesh (6 σ).”.

- Figure 2G and Figure 3C – The surface-rendered model of the outward-facing (Figure 2) and inward-facing (Figure 3) subunit from the HsZnT8-WT structure obtained in the absence of Zn. It would make better sense to show both conformations side-by-side in the same figure.

As suggested, we have shown surface-rendered models for both outward and inward conformations side-by-side in Figure 3C.

- Figure 3—figure supplement 1B. The ResMap data could be presented for additional clarity by using an alternate coloring scheme or the same scale as in Figure 1—figure supplement 2.

We have changed the coloring scheme for the local resolution distribution result in Figure 3—figure supplement 1B.

- While the % particles in each image processing step used to determine the final map were included as previously requested, the % particles in the discarded maps from 3D classification and 3D refinement will provide additional clarity for data distribution.

We have added the % particles for all the 3D classes.

- Indicate if RMSDs are based on C-α or all-atom.

The RMSDs were calculated based on C-α atom only. We have indicated this in the figure legend.